# Evaluating Acylsugars-Mediated Resistance in Tomato against *Bemisia tabaci* and Transmission of Tomato Yellow Leaf Curl Virus

**DOI:** 10.3390/insects11120842

**Published:** 2020-11-28

**Authors:** Wendy G. Marchant, Saioa Legarrea, John R. Smeda, Martha A. Mutschler, Rajagopalbabu Srinivasan

**Affiliations:** 1Department of Entomology, University of Georgia, 2360 Rainwater Road, Tifton, GA 31793, USA; wgmarch@att.net (W.G.M.); saioalegarrea@gmail.com (S.L.); 2Section of Plant Breeding and Genetics, School of Integrative Plant Science, Cornell University, 257 Emerson Hall, Ithaca, NY 30602, USA; smeda.john@ufl.edu (J.R.S.); mam13@cornell.edu (M.A.M.); 3Department of Entomology, University of Georgia, 1109 Experiment Street, Griffin, GA 310223, USA

**Keywords:** *Bemisia tabaci*, begomovirus, transmission, host resistance, secondary metabolites, management

## Abstract

**Simple Summary:**

The sweetpotato whitefly, *Bemisia tabaci*, besides causing feeding-related injuries, also transmits the economically devastating tomato yellow leaf curl virus (TYLCV) to tomato plants. The main management options include host resistance and insecticides. Host resistance is imparted against TYLCV and not against the vector. Incorporating host resistance against whiteflies could broaden management options. Acylsugars are secondary metabolites exuded from trichomes of wild solanums that can negatively impact numerous herbivores, including whiteflies. This study examined the effects of acylsugar-producing tomato lines with fatty acid quantitative trait loci introgressions from *Solanum pennellii* LA716 into cultivated tomato on whitefly preference and fitness, and whitefly-mediated TYLCV inoculation and subsequent TYLCV acquisition. Overall, the acylsugar-producing lines negatively affected whitefly preference and fitness in comparison with the non-acylsugar hybrid. Acylsugars’ mediated antixenosis and antibiosis effects against whiteflies were documented. Acylsugar-producing lines also reduced whitefly-mediated TYLCV inoculation and subsequent TYLCV acquisition by whiteflies in comparison with the non-acylsugar hybrid. These results suggest that acylsugar-mediated resistance against whiteflies could complement already existing TYLCV resistance in tomato cultivars/hybrids and could reduce the heavy reliance on chemical control of whiteflies.

**Abstract:**

The sweetpotato whitefly, *Bemisia tabaci*, is a major pest of cultivated tomato. Whitefly feeding-related injuries and transmission of viruses including tomato yellow leaf curl virus (TYLCV) cause serious losses. Management strategy includes planting resistant cultivars/hybrids. However, TYLCV resistance is incomplete and whiteflies on TYLCV-resistant cultivars/hybrids are managed by insecticides. Acylsugars’-mediated resistance against whiteflies has been introgressed from wild solanums into cultivated tomato. This study evaluated acylsugar-producing tomato lines with quantitative trait loci (QTL) containing introgressions from *Solanum pennellii* LA716, known to alter acylsugars’ levels or chemistry. Evaluated acylsugar-producing lines were the benchmark line CU071026, QTL6/CU071026—a CU071026 sister line with QTL6, and three other CU071026 sister lines with varying QTLs—FA2/CU71026, FA7/CU071026, and FA2/FA7/CU071026. Non-acylsugar tomato hybrid Florida 47 (FL47) was also evaluated. Acylsugars’ amounts in FA7/CU071026 and FA2/FA7/CU071026 were 1.4 to 2.2 times greater than in other acylsugar-producing lines. Short chain fatty acid, i-C5, was dominant in all acylsugar-producing lines. Long chain fatty acids, n-C10 and n-C12, were more abundant in FA7/CU071026 and FA2/FA7/CU071026 than in other acylsugar-producing lines. Whiteflies preferentially settled on non-acylsugar hybrid FL47 leaves over three out of five acylsugar-producing lines, and whiteflies settled 5 to 85 times more on abaxial than adaxial leaf surface of FL47 than on acylsugar-producing lines. Whiteflies’ survival was 1.5 to 1.9 times lower on acylsugar-producing lines than in FL47. Nevertheless, whiteflies’ developmental time was up to 12.5% shorter on acylsugar-producing lines than on FL47. TYLCV infection following whitefly-mediated transmission to acylsugar-producing lines was 1.4 to 2.8 times lower than FL47, and TYLCV acquisition by whiteflies from acylsugar-producing lines was up to 77% lower than from FL47. However, TYLCV accumulation in acylsugar-producing lines following infection and TYLCV loads in whiteflies upon acquisition from acylsugar-producing lines were not different from FL47. Combining TYLCV resistance with acylsugars’-mediated whitefly resistance in cultivated tomato could substantially benefit whiteflies and TYLCV management.

## 1. Introduction

The sweetpotato whitefly, *Bemisia tabaci* (Gennadius), is an insect pest of enormous economic importance to many vegetable crops including tomato, *Solanum lycopersicum* L. [1,2,3,4]. *Bemisia tabaci* is a cryptic species complex that comprises morphologically identical sibling species [5,6,7,8,9]. Two of the cryptic species, MEAM1 and MED, are highly invasive and have displaced other native cryptic species in many areas of the world [7,8,9,10,11]. The Middle East Asia Minor 1 (MEAM 1—previously biotype B) is the most prevalent invasive cryptic species in the United States [12]. Feeding by *B. tabaci* MEAM1 causes phytotoxic effects such as silverleaf and white stem in squash plants, and irregular ripening of tomato fruits [12,13]. Honeydew produced by whiteflies can result in sooty mold growth and suppress photosynthetic potential [14]. Even more damaging is the ability of *B. tabaci* to transmit viruses to crops. *Bemisia tabaci* can transmit plant viruses belonging to *Closteroviridae*, *Potyviridae*, *Secoviridae*, *Betaflexiviridae*, and most importantly, *Geminiviridae* [15,16]. In tomato, *B. tabaci* transmits tomato yellow leaf curl virus (TYLCV), which can cause substantial yield loss [17].

TYLCV originated in the Middle East and has rapidly spread around the world within the past half-century [18,19]. Symptoms of TYLCV on tomato plants include curling of leaves, chlorosis, stunted growth, and reduced fruit yield. Unprotected fields can result in 100% incidence and yield loss [20,21]. One of the effective tactics to mitigate TYLCV-induced yield losses is to plant TYLCV-resistant tomato cultivars/hybrids [22,23,24]. TYLCV resistance is conferred by one or a few semi-dominant *Ty* genes [25,26,27,28,29,30]. Upon infection, TYLCV-resistant cultivars/hybrids do not exhibit hypersensitive response and can get systemically infected, they display milder TYLCV symptoms than susceptible cultivars/hybrids [31]. TYLCV also replicates in TYLCV-resistant cultivars/hybrids, albeit at a reduced level than in susceptible cultivars/hybrids [23,32]. Therefore, TYLCV-resistant cultivars/hybrids cannot be used as a stand-alone management tactic, and insecticides are often used to manage whiteflies on resistant cultivars/hybrids [20]. Routine insecticide usage raises concerns about insecticide resistance development by whiteflies [33,34,35,36]. Given the scenario, tomato cultivars/hybrids with resistance to both the whitefly vector and the virus would be ideal. Tomato genotypes with acylsugars’ introgressed from wild solanums are known to confer resistance to whiteflies and are being used in tomato breeding [37,38,39,40,41,42].

Acylsugars are esterified monosaccharides (glucose) or disaccharides (sucrose) with straight or branched fatty acid side chains of varying lengths determined by the number of carbon atoms [43]. Several genera of the *Solanaceae* are known to produce acylsugars [39,43,44,45,46,47]. They are exuded by a specific type of glandular trichome. A total of eight different types of trichomes that have been identified in tomato [48,49]. The type IV trichome is known to exude acylsugars. Type IV trichome has glandular cells at its tip and is capable of synthesizing and exuding acylsugars that deter insect feeding and/or oviposition [37,50,51,52,53]. Cultivated tomato plants can also produce acylsugars, but at very low levels and not as droplets, with almost no effects on herbivores. Genes/QTL impacting acylsugar production have been introgressed from wild relatives such as *Solanum pimpinellifolium* L., *Solanum pennellii* Correll, and *Solanum galapagense* (S.C.Darwin and Peralta) into cultivated tomato [38,40,41,42,54,55]. Resulting acylsugar-producing genotypes have been evaluated against a number of different tomato herbivores such as aphids, lepidopterans, thrips, and whiteflies [39,47,56,57,58,59,60,61].

Acylsugars’-mediated antibiosis and antixenosis effects against whiteflies in tomato genotypes have been documented before. For instance, application of purified acylsugars of *S. pennellii* LA716 altered the settling and reduced the oviposition of whiteflies [37,39]. Fewer eggs and nymphs were found on the lower leaves of acylsugar-producing than non-acylsugar producing genotypes [37,38,39,55,56,62,63]. Whiteflies also landed and oviposited less frequently on acylsugar-producing lines compared with a non-acylsugar cultivar, but such effects were apparent at the 10-leaf stage and only beyond a threshold concentration of acylsugars [63,64].

While a few studies have reported the effects of acylsugars-mediated fitness effects on whiteflies, relatively fewer have examined how these fitness effects on whiteflies interfere with their acquisition and inoculation abilities of tomato-infecting persistently-transmitted viruses such as TYLCV. An earlier study examined the effects of acylsugar mediated resistance introgressed from *S. pimpinellifolium* line TO-937 into cultivated tomato on TYLCV transmission [58]. The line with acylsugars ABL14-8 was infected with TYLCV at a reduced percentage in comparison with the cultivar without acylsugars; however, acylsugars did not completely hinder TYLCV inoculation by whiteflies. Secondary spread of TYLCV following acquisition by whiteflies from TYLCV-infected acylsugar-producing ABL14-8 and inoculation of the non-acylsugar producing cultivar was also reduced but not eliminated [58].

Acylsugars’ pertaining QTLs have also been introgressed from *S. pennellii* LA716 into cultivated tomato, resulting in the Cornell University breeding program benchmark line CU071026. This initial line produced 15% of acylsucrose in comparison with *S. pennellii* and had a genome 11%, which is *ca. S. pennellii* genome [38]. Purified acylsugars with different chemotypes from CU070126 and other *S. pennellii* accessions revealed that acylsugars with different fatty acid structures had differential and synergistic impacts in reducing oviposition by MEAM1 whiteflies [39]. This indicated that breeding for acylsugar chemistry could be as important as for acylsugar level. QTLs that affect the type of fatty acid or sugar found in acylsugars were identified and transferred into CU017026 to create sister lines with different acylsugar chemistries [38,40,41,42,65].

In this study, interactions of whiteflies with five acylsugar genotypes including the benchmark CU071026 line and four other sister lines, FA2/CU071026, FA7/CU071026, FA2/FA7/CU071026, and QTL6/CU071026, were characterized. For this purpose, whitefly preference and fitness on acylsugar-producing lines in comparison with FL47 were evaluated using two-way settling assays. FL47 is a non-acylsugar tomato hybrid, and it is not resistant to whiteflies and/or TYLCV. In addition to effects on whiteflies’ fitness, the ability of whiteflies to inoculate TYLCV to acylsugar-producing lines was assessed both qualitatively and quantitatively via endpoint PCR and quantitative PCR, respectively. Further, the ability of whiteflies to acquire TYLCV from acylsugar-producing lines in comparison with the non-acylsugar tomato hybrid was also assessed qualitatively and quantitatively. The goal was to determine whether the acylsugar-producing lines would suppress the primary spread of TYLCV by limiting whitefly-mediated inoculation of TYLCV and/or TYLCV accumulation in the acylsugar-producing lines, and subsequently whether the secondary spread of TYLCV would be reduced by limiting whitefly acquisition of TYLCV and/or TYLCV accumulation in whiteflies.

## 2. Materials and Methods

### 2.1. Tomato Acylsugar-Producing Lines

Origin of CU071026 sister lines: a line developed in parallel with CU071026 possessed a QTL on chromosome 6, which raised acylsugar level and also trichome density. The chromosome 6 introgression containing QTL6 was transferred to CU071026 creating a sister line QTL6/CU071026 [38]. QTL controlling fatty acid constituents of acylsugars were identified [65], and mono-introgression lines with each containing one of the QTLs affecting acylsugar fatty acids were crossed and backcrossed to CU071026 to create three sister lines—FA2/CU071026, FA7/CU071026, and FA2/FA7/CU071026 [40,41,42].

### 2.2. Whiteflies and TYLCV

The sweetpotato whitefly, *B. tabaci* cryptic species Middle East-Asia Minor 1 (MEAM1) (GenBank accession number MN970031-32) has been maintained in a greenhouse since its first collection in 2009 from Tifton, Georgia, GA, USA. In the greenhouse, the whiteflies were maintained on 15 to 20 cm tall cotton plants in whitefly-proof cages at 25–30 °C and a 14 h L:10 h D photoperiod. The TYLCV isolate (GenBank accession number KY965880) first collected from a TYLCV-infected tomato plant in 2009 in Montezuma, Georgia, USA has since been maintained through whitefly-mediated transmission in the greenhouse at conditions stated above in the tomato cultivar FL47 (Seminis Vegetable Seeds, St. Louis, MO, USA).

### 2.3. Acylsugars’ and Fatty Acid Profiles

A total of seven foliar samples were used from each acylsugar-producing line for estimation of total acylsugars and fatty acid profiles. Total acylsugar contents were assessed from foliar tissue of ~10-week-old tomato plants of each line. Two lateral leaflet samples were collected from each plant, placed in plastic scintillation vials, and then fully dried. The dried leaflet samples were each washed with 3 mL of methanol, and acylsugar levels in the rinsate were measured on aliquots of the rinsate by the method described by Leckie et al. [38]. The remnant acylsugar rinsate was also used for estimation of fatty acid side chain profiles as described by Leckie et al. [43]. Succinctly, acylsugar rinsates were individually dried and resuspended in methanol along with the internal standard methyl heptanoate and subjected to GC/MS analyses. Component fatty acid peaks were identified and quantified in reference to the internal standard using mass spectrometry.

### 2.4. Whitefly-Mediated Inoculation of TYLCV to Acylsugar-Producing Lines

Plants of five acylsugar-producing tomato lines and FL47 were used for whitefly-mediated inoculation of TYLCV. Plants of each line and FL47 were grown in the greenhouse in whitefly-proof cages as stated above until they reached the ten true-leaf stage. A total of twenty viruliferous whiteflies after a 72 h acquisition access period (AAP) on TYLCV-infected FL47 plants were clip-caged to a leaf and provided with an inoculation access period (IAP) for 24 h. Whiteflies were removed and plants were sprayed with insecticidal soap (Garden Safe, Bridgeton, MO, USA) to kill remaining adults or eggs laid on the plants. A total of six plants of each acylsugar line and FL47 were inoculated per experiment. The experiment was conducted four times (n = 24). Plants were maintained for three weeks to allow development of infection. Tissue samples from topmost fully expanded leaves were collected and washed to remove external contamination. One hundred milligrams of leaf tissue from each plant was ground in 1.5 mL microcentrifuge tube with a plastic pestle. Total DNA was extracted by using the GeneJET Plant Genomic DNA Purification Kit as per the manufacturer’s protocol (Thermo Scientific, Waltham, MA, USA). PCR was then conducted to determine TYLCV infection status. Primers used were C2-1201 (5′-CATGATCCACTGCTCTGATTACA-3′) and C2-1800V2 (5′-TCATTGATGACGTAGACCCG-3′); they targeted a 695-nucleotide region of the TYLCV genome that encompassed the entire C2 gene. The PCR reactions were run in 10 μL reactions with 5 μL of GoTaq^®^ Green Master Mix (Promega Corporation, Madison, WI, USA), 2 μL of water, 0.5 μL of each primer at 10 μM concentration, and 2 μL of DNA extract. The PCR program had an initial denaturation step at 94 °C for 2 min followed by 30 cycles of 94 °C for 30 s, 52 °C for 30 s, 72 °C for 1 min, and a final extension at 72 °C for 5 min. Percent infection data were analyzed in SAS 9.4 (SAS Institute Inc., Cary, NC, USA) using the GLIMMIX PROCEDURE assuming a binomial distribution. Treatments were considered as fixed effects and experiments and replications were considered random effects. Treatment differences were identified using the Type III sum of squares statistics, and treatment means were separated using the Tukey–Kramer method.

Samples that tested positive were then subjected to real-time PCR to quantitate TYLCV load relative to the tomato 25S rRNA gene using the mathematical formula from Pfaffl [66]. Primers used for quantitative PCR were TYLC-C2-For (5′-GCAGTGATGAGTTCCCCTGT-3′) and TYLC-C2-Rev (5′-CCAATAAGGCGTAAGCGTGT-3′), which covered a 102-nucleotide region over the TYLCV C2 gene. The real-time PCR reactions were run in 25 μL reactions with 12.5 μL of GoTaq^®^ qPCR Master Mix (Promega Corporation, Madison, WI), 6.5 μL of water, 0.5 μL of each primer at 10 μM concentration, and 5 μL of DNA extract. The real-time PCR program for the C2 gene had an initial denaturation step at 95 °C for 2 min followed by 40 cycles of 95 °C for 15 s and 60 °C for 1 min, followed by melting curve analysis. Primers used for the amplification of the tomato 25S rRNA gene were Tomato 25S rRNA F (5′-ATAACCGCATCAGGTCTCCA-3′) and Tomato 25S rRNA R (5′-CCGAAGTTACGGATCCATTT-3′) [67]. The real-time PCR program for the tomato 25S rRNA gene had an initial denaturation step at 95 °C for 2 min followed by 40 cycles of 95 °C for 15 s and 53 °C for 1 min followed by melting curve analysis. All samples were included in duplicates. Differences in TYLCV loads in leaves of acylsugar-producing lines in comparison with FL47 were analyzed using the GLIMMIX Procedure in SAS assuming a Gaussian distribution. Treatments were considered as fixed effects and experiments and replications were considered random effects. Treatment differences were identified using the Type III sum of squares statistics, and treatment means were separated using the Tukey–Kramer method.

### 2.5. Whitefly Acquisition of TYLCV from TYLCV-Infected Acylsugar-Producing Tomato Lines

A total of twenty 48-h-old adult whiteflies were clip-caged to a leaflet of the tenth true-leaf of an infected tomato plant that had been inoculated with TYLCV four weeks earlier. Whiteflies were clip-caged for 72 h and removed. A total of six whiteflies were sampled per clip-cage. The experiment was conducted three times (n = 18). DNA extraction was performed on individual whiteflies using Instagene Matrix (BioRad, Hercules, CA, USA) using manufacturer’s instructions. Samples were tested for TYLCV using primers C2-1201 and C2-1800V2. Percent acquisition data were analyzed in SAS using the GLIMMIX Procedure and assuming a binomial distribution as described above.

Whitefly samples that tested positive for TYLCV with endpoint PCR were subjected to quantitative PCR with the TYLC-C2-For and TYLC-C2-Rev primer pair. Values were normalized with the whitefly β-actin gene, which was amplified with the primer pair whitefly β-actin F (5’–TCTTCCAGCCATCCTTCTTG–3’) and whitefly β-actin R (5’–CGGTGATTTCCTTCTGCATT–3’) [68]. The quantitative PCR program had an initial 95 °C denaturation step for 2 min, followed by 40 cycles of 95 °C for 15 s and 60 °C for 1 min followed by melting curve analyses. Values for TYLCV and the whitefly β-actin were used in the equation developed by Pfaffl [66] for relative quantitation of TYLCV DNA. All samples were included in duplicates. TYLCV load in whiteflies following acquisition from acylsugar-producing lines and FL47 was analyzed using the GLIMMIX procedure in SAS as stated earlier.

### 2.6. Whitefly Settling Assays

Acylsugar tomato lines CU071026, FA2/CU071026, FA7/CU071026, FA2/FA7/CU071026, and QTL6/CU071026 were each paired with FL47 in a settling arena. Additionally, the lines FA2/CU071026, FA7/CU071026, QTL6/ CU071026, and FA2/FA7/ CU071026 were each paired with CU071026. Settling assays were conducted using genotype pairs. Each experiment was conducted twice using six plants per line each time (n = 12 for each settling assay pair/combination). One leaf from each ten true-leaf stage tomato plant was inserted into a settling arena. A vial containing one hundred whiteflies was placed at the bottom of the arena [32]. After 24 h, the numbers of whiteflies settling on both the abaxial and adaxial side of each leaf were tabulated. Preference data were analyzed with a two-way ANOVA using the GLIMMIX Procedure in SAS (SAS Institute, Cary, NC, USA). The lines and leaf surfaces were considered as fixed effects, whereas experiments and replications were considered as random effects. Settling differences between treatment pairs and leaf surfaces were assessed using Type III statistics.

### 2.7. Whitefly Survival from Eggs to Late Instar Immatures

The percentage of eggs surviving to become late- (third or fourth) instar nymphs was monitored on acylsugar-producing lines CU071026, QTL6/CU071026, FA2/FA7/CU071026, and FL47. Adult female whiteflies were clip-caged on the leaf of a tomato plant and allowed to lay eggs for two days. A total of six plants were used per line and the experiment was conducted twice (n = 12). Female whiteflies were then removed, and the number of eggs was counted. Egg-bearing plants were maintained for two weeks in insect proof cages placed in the greenhouse maintained under the conditions described above. Two weeks later, the number of third or fourth instar nymphs were recorded in each clip cage. Percentage survival data were analyzed using the GLIMMIX procedure in SAS as described before.

### 2.8. Whitefly Developmental Time from Egg to Adult Eclosion

Whitefly developmental time from egg to adult was measured on acylsugar-producing lines CU071026, QTL6/CU071026, FA2/FA7/CU071026, and FL47. Adult female whiteflies were clip-caged to a fully expanded top leaf of a tomato plant and allowed to lay eggs for two days. A total of six plants were used per line and the experiment was conducted twice (n = 12 for each treatment). Eggs were monitored daily until adult eclosion. The time from egg to adult eclosion was recorded. A median one-way analysis of variance was conducted using the NPARIWAY Procedure in SAS to assess differences in developmental time among acylsugar-producing lines.

## 3. Results

### 3.1. Acylsugars’ and Fatty Acid Profiles

Total acylsugars’ content varied with the lines evaluated (Figure 1) (F_(4,52)_ = 11.81; *p* < 0.0001). The amount of acylsugars present in CU071026, FA2/CU071026, and QTL6 were not different from each other. However, the amount of acylsugars in FA7/CU071026 and FA2/FA7/CU071026 were greater than in CU071026, FA2/CU071026, and QTL6/CU071026.

The fatty acid profile also varied with the acylsugar-producing tomato line evaluated (Table 1; Appendix A). A total of fifteen fatty acids were identified. The most dominant fatty acid in all the lines appeared to be i-C5 (3-methylbutanoate iso-branched 5-carbon acyl group). The percentage of i-C5 (3-methylbutanoate iso-branched 5-carbon acyl group) varied from 45 to 60%. The other five top fatty acids included ai-C5 (2-methylbutanoate anteiso-branched 5-carbon acyl group), i-C11 (9-methylbdecanoate iso-branched 11-carbon acyl group), n-C12 (n-dodecanoate straight chain 12-carbon acyl group), i-C4 (2-methylproponate iso-branched 4-carbon acyl group), and n-C10 (n-decanoate straight chain 10-carbon acyl group). These six fatty acids accounted for more than 95% of the total in each acylsugar-producing line evaluated. The dominant fatty acid i-C5 percentage was lower in FA7/CU071026 and FA2/FA7/CU071026 than in other lines. On the contrary, percentages of n-C10 and n-C12 were greater in FA7/CU071026 and FA2/FA7/CU071026 in comparison with other acylsugar-producing lines.

### 3.2. Whitefly-Mediated Inoculation of TYLCV to Acylsugar-Producing Lines

The TYLCV infection percentages following whitefly-mediated inoculation did not vary within acylsugar-producing tomato lines, but the infection percentages were significantly lower in acylsugar-producing lines in comparison with FL47 except for QTL6/CU071026 (F_(5,138)_ = 3.23, *p* = 0.0412) (Figure 2a).

Despite the reduced infection percentage in acylsugar-producing tomato lines, TYLCV loads among different acylsugar-producing tomato lines did not differ from that of FL47 (F_(5,61)_ = 0.5522, *p* = 0.7360) (Figure 2b).

### 3.3. Whitefly Acquisition of TYLCV from TYLCV-Infected Acylsugar-Producing Tomato Lines

The number of whiteflies that acquired TYLCV from various tomato lines differed from that of FL47 (F_(5,103)_ = 3.99, *p* = 0.0085). TYLCV acquisition by whiteflies from QTL6 was lower in comparison with FL47, and it was not different from other acylsugar-producing lines (Figure 3a). Additionally, the TYLCV load acquired by viruliferous whiteflies, as determined by quantitative PCR, varied between acylsugar-producing tomato lines and FL47 (F_(5,45)_ = 4.31, *p* = 0.0052) (Figure 3b). TYLCV load acquired from FA2/FA7/CU071026 was higher in comparison with FL47. The TYLCV load acquired from FL47 did not vary from any other acylsugar-producing line.

### 3.4. Whitefly Settling Assays

Over two-thirds of the released whiteflies settled on treatment leaves. The whitefly settling percentages ranged from 66.57 to 90.57% across treatment pairs. Whiteflies’ settling was affected both by the presence of acylsugars in tomato lines and by the leaf surface. Whiteflies preferentially settled on the non-acylsugar hybrid FL47 over three acylsugar-producing lines (CU071026, FA7/CU071026, and QTL6/CU071026) (Figure 4a–e; Appendix A). Whiteflies settling was significantly higher on the abaxial versus the adaxial leaf surface in the non-acylsugar hybrid FL47 compared with all five acylsugar-producing lines.

The whitefly settling percentages in assays involving acylsugar-producing lines alone ranged from 49.58 to 75.49% across treatment pairs. Whiteflies did not exhibit a preference for CU071026 when compared with four other acylsugar-producing lines (Figure 5a–d; Appendix A). On acylsugar-producing lines, whiteflies did not exhibit any preference for the abaxial or adaxial leaf surface in two instances (Cu071026 vs FA2/CU071026 and Cu071026 vs QTL6/CU071026). In one instance, whiteflies preferentially settled on the adaxial side of FA7/CU071026 leaves when compared with Cu071026. In another instance, whiteflies preferentially settled on the abaxial side leaves of FA2/FA7/CU071026 when compared with Cu071026 (Figure 5). Overall, settling on abaxial leaf surfaces of all acylsugar-producing lines was much lower when compared with FL47.

### 3.5. Whitefly Survival from Egg to Late Instar Immatures

The percentage of insects that survived from the egg stage into late (third or fourth) instar nymphal stage two weeks post oviposition was higher on the non-acylsugar hybrid FL47 compared with acylsugar-producing lines CU071026, FA2/FA7/CU071026, and QTL6/CU071026 (Table 2). 

### 3.6. Whitefly Developmental Time from Egg to Adult Eclosion

The developmental time from egg to adult eclosion in individual insects varied between FL47 and acylsugar-producing lines (Χ^2^ = 24.4499, Χ^2^ < 0.0001). The whitefly developmental time on FL47 differed from the three acylsugar-producing lines CU071026, FA2/FA7/CU071026, and QTL6/CU071026. Whiteflies took two to three days longer to develop on FL47 than on acylsugar-producing lines (Table 3).

## 4. Discussion

Acylsugars exuded by glandular trichomes have been known to negatively affect the fitness of whiteflies [37,38,39,64]. Breeding programs have attempted to incorporate acylsugars-mediated resistance from several wild solanum species against whiteflies [38,40,41,42,64,69]. Transfer of select additional *S. pennellii* LA716 introgressions from mono-introgression lines [70,71] into the Cornell University acylsugar breeding line CU071026 led to the development of acylsucrose producing lines with alterations in acylsugar fatty acid side chains used in this study [40,41]. The current study aimed to characterize interactions between whiteflies and five acylsugar-producing lines in comparison with a non-acylsugar tomato hybrid. Acylsugars’ accumulation in FA7/CU071026 and FA2/FA7/CU071026 were almost twice as high than that of CU071026. Similar results were observed when CU071026 was compared with other backcross lines [38,60,61,65]. The short chain fatty acids such as i-C5 and ai-C5 (3-methylbutanoate iso-branched 5-carbon acyl group and 3-methylbutanoate anteiso-branched 5-carbon acyl group) seem to be the most predominant in all the lines, and short-chained fatty acids accumulation levels were at least two times more than that of the long-chained fatty acids i-C11 and nC-12. Proportions of fatty acids also varied among lines, significantly in a few instances. Results provided evidence for acylsugars-induced antixenosis effects on whiteflies by affecting their settling preference, especially on the abaxial leaf surface, and direct antibiosis-induced effects such as reduced whitefly survival and development on acylsugar-producing lines in comparison with the non-acylsugar hybrid. The acylsugar-producing lines with both altered fatty acid profiles and increased acylsugar amounts compared with CU071026, reduced the whitefly preference and enhanced the magnitude of whitefly suppression when compared with the control FL47. Since the lines have both increased acylsugar amounts and altered fatty acid profiles, the data were not sufficient to determine whether one, the other, or both of these changes combined are responsible for the greater whitefly suppression. While *B. tabaci* feeding injuries can inflict serious losses, its ability to transmit viruses can be devastating [4]. The currently available TYLCV-resistant cultivars/hybrids only exhibit moderate resistance to TYLCV and are not resistant/tolerant to whiteflies [23,31]. This study also attempted to qualitatively and quantitively characterize interactions between the acylsugar-producing lines and TYLCV. Results indicated that the presence of acylsugars suppressed TYLCV transmission by almost three times compared with the non-acylsugar tomato hybrid. However, once infected, the acylsugar-producing lines accumulated viral loads similar to the non-acylsugar tomato hybrid with exceptions. Consequently, three-to-four times fewer whiteflies acquired TYLCV from TYLCV-infected acylsugar-producing lines in comparison with the non-acylsugar hybrid with an exception, but upon acquisition from acylsugar-producing lines, the viral loads were often not different from the non-acylsugar hybrid. Reduced whitefly survival and development on acylsugar-producing lines in comparison with the non-acylsugar hybrid could also influence TYLCV spread and management. Overall, acylsugar-producing lines have the potential to reduce the primary and secondary spread of TYLCV in tomato.

Antixenosis-based effects such as reduced ovisposition and virus transmission involving CU071026 and acylsugar-producing lines with *S. pennellii* LA716 introgressions on phytovirus transmitting vectors such as thrips and whiteflies have been documented before [38,41,60,61]. Oviposition reduction, though a useful trait, still might not limit the primary spread of the virus, as viruliferous vectors including whiteflies and thrips can transmit the virus within a few minutes of feeding [17,72]. This study focused on preference of whiteflies to assess if any reduced preference could lead to reduced TYLCV transmission. Whiteflies, when given a choice, preferred to settle on the non-acylsugar hybrid over acylsugar-producing lines. When the choice was between two acylsugar-producing lines, no preferential settling was observed. However, whiteflies exhibited a clear preference to settle on the abaxial side of the leaf on FL47. On the acylsugar-producing lines, whiteflies typically showed no preference for either the abaxial or adaxial side. This effect was also documented by Rodríguez-López et al. [64], as whiteflies preferred the abaxial leaf surface in their control non-acylsugar cultivar, Moneymaker, but showed no preference toward abaxial or adaxial leaf surfaces in the acylsugar-producing line, ABL14-8. Settling on adaxial surface of leaves could predispose whiteflies to natural enemies, insecticide sprays, and adverse weather conditions and could limit their ability to successfully utilize and colonize hosts [73]. Electrical Penetration Graph observations demonstrated that whiteflies most often fed on the abaxial leaf surface in the non-acylsugar cultivar Moneymaker, but with the acylsugar-producing line, whiteflies failed to feed on the abaxial leaf surface and only did so on the adaxial leaf surface [64]. In a study examining the effects of acylsugars on the two-spotted spider mite, Rakha et al. [74] observed that in *S. pimpinellifolium* line VI030462, type IV trichomes were found in higher abundance on the abaxial leaf surface than the adaxial leaf surface. The two-spotted spider mites preferred to lay their eggs at significantly higher levels on the adaxial leaf surface compared with abaxial leaf surface of this line. Type IV glandular trichomes were also found on the acylsugar-producing tomato line ABL 10-4 than on the non-acylsugar cultivar and negatively impacted the fitness of the greenhouse whitefly, *Trialeurodes vaporariorum* Westwood [59]. It is possible that, in the current study, the type IV trichomes could have been at a higher density on the abaxial than adaxial leaf surface of acylsugar-producing lines, and is yet to be determined. Despite the general lack of preference for acylsugar-producing lines, whitefly settling on acylsugar-producing lines was not completely absent. Whiteflies can still feed on these lines and inoculate the virus, thereby initiating the primary spread of the virus.

Survival percentages of whiteflies developing from egg to third or fourth instar nymphal stage were also lower for on the abaxial leaf surface of acylsugar-producing lines compared with FL47. Interestingly, whiteflies took longer to develop on control hybrid FL47 than on acylsugar-producing genotypes. The quicker developmental time might be an adaptive strategy to effectively utilize a non-preferred host. Such a phenomenon has been documented in thrips and other insects but not in whiteflies [75,76]. These direct effects on the whitefly fitness provide evidence for antibiosis-induced resistance. Similar results were obtained by Leckie et al. [39], where whitefly oviposition and development on the benchmark line CU071026 and others (not presented in this study) were suppressed. Another controlled field study conducted revealed that very few whitefly adults and immatures were found on some acylsugar-producing lines in relation to a non-acylsugar cultivar under field conditions [65]. These consistent results across various CU071026-*S. pennellii* LA716 derived lines and across studies indicate that negative fitness effects on whiteflies could suppress whitefly populations, reduce the acquisition of viruses from these lines, and limit the secondary spread of the virus.

Whiteflies, when clip-caged on leaves of acylsugar-producing lines, inoculated the virus less efficiently than on the non-acylsugar hybrid FL47. The whiteflies were clip-caged on the abaxial leaf surface, and it is possible that glandular trichome exuded acylsugars could have deterred the whiteflies from feeding on the leaves. Due to this deterrence, whiteflies might have not been able to reach the phloem or engage in sustained phloem feeding. Phloem feeding is essential to inoculate TYLCV, as it is phloem limited. An electrical penetration graph (EPG) study by Rodríguez-López et al. [64] noted that whiteflies took longer to initiate first probing and probed less on the acylsugar-producing line ABL 14-8 than on non-acylsugar cultivar moneymaker; however, the duration of ingestion was not different between the acylsugar-producing line and control cultivar Moneymaker once feeding was initiated. These studies reiterate that acylsugars could still permit whitefly feeding, and with increased densities of whiteflies as often seen in the southeastern United States, their efficacy to suppress TYLCV infection remains to be evaluated. Nevertheless, the virus infection reduction seen in this study is encouraging. Another concern remains that in some lines acylsugars begin to be expressed only at the eight-to-ten leaf stage, and prior to that they could still be susceptible to whiteflies and whitefly-mediated inoculation of TYLCV [58,63]. However, the production of acylsugars in CU070126 and derived lines seem to occur at younger leaf stages, thereby providing protection earlier than other acylsugar-producing lines.

Accumulation of TYLCV loads in acylsugar-producing lines did not differ from the non-acylsugar cultivar. This outcome was partly expected, considering that the acylsugar-producing lines do not possess any TYLCV resistance conferring genes and acylsugars are present exclusively on the outer surface of leaves, acylsugars might not impact virus after its inoculation into the phloem tissue. TYLCV resistance is conferred by *Ty* genes (1 through 6), and the resistance is characterized by reduced symptom expression and subdued virus replication leading to reduced virus load in comparison with the tomato cultivars or hybrids without the *Ty* genes [23,32]. None of the acylsugar-producing lines evaluated in this study possessed a *Ty* gene. Therefore, when acylsugar-producing lines are infected, they could permit unimpeded virus replication following inoculation of TYLCV into the phloem. Consequently, whiteflies were able to acquire similar amounts of TYLCV following an AAP on TYLCV-infected acylsugar-producing lines in comparison with a TYLCV-infected FL47 with one exception (FA7/CU071026). Whiteflies have been known to acquire TYLCV in a density dependent fashion from their host plants [23,32], and it is not a surprise that infected plants of the acylsugar-producing lines seem to function as effective inoculum sources on par with FL47; the acylsugar-producing lines can be effective at deterring the vector and reducing TYLCV inoculation, but not TYLCV replication post infection.

Overall, acylsugar-producing lines examined in this study seem to negatively impact whitefly preference and whitefly fitness via both antixenosis and antibiosis. While the acylsugar-producing lines have been evaluated several times, this study focused on how reduction of whitefly settling and suppression of whitefly populations could impact TYLCV inoculation and TYLCV spread. Results indicated that the negative impact induced by acylsugars on whiteflies reduced whitefly-mediated inoculation. Additionally, whitefly acquisition of TYLCV was reduced from acylsugar-producing lines in comparison with a non-acylsugar hybrid. These results suggest that acylsugar-mediated resistance against whiteflies could result in reduced or delayed primary spread of the virus and subsequently translate to reduced or delayed secondary spread of the virus. Even a delay in the virus spread without reduction could still be useful, as the susceptibility to TYLCV decreases with plant age, and ensuing mature plant resistance can mitigate yield losses. While the results from the current study are encouraging, they should be cautiously interpreted. In the current study, only twenty viruliferous whiteflies were used to inoculate a single tomato plant, under field conditions substantially more whiteflies can be observed on a tomato plant. Under such intense whitefly pressure on tomato plants, the efficacy of acylsugar-producing lines on TYLCV epidemics remains to be evaluated.

## 5. Conclusions

Whiteflies and whitefly-transmitted TYLCV are arguably the most important issues affecting tomato production in many parts of the world. Their current management relies on planting TYLCV resistant cultivars/hybrids with *Ty* genes. These cultivars/hybrids are not resistant to whiteflies, and whitefly management on these resistant genotypes is typically accomplished by insecticide applications. These applications can drive up production costs and are typically not environmentally friendly. On the contrary, acylsugars-mediated resistance can suppress whitefly populations, but do not possess any TYLCV resistance. While neither of these options alone provides sufficient control, combining whitefly resistance with TYLCV resistance could be ideal. There certainly can be difficulties and tradeoffs associated with introgressing multiple QTL or resistance genes from multiple wild species into cultivated tomato; however, given the advancements in marker assisted selection, improvements in annotating the tomato genome, and the availability of thousands of mapped markers, this is becoming far easier. In fact, the QTL6/CU071026 line was created in two forms, one with and the other without the *Sw-5* gene for resistance against tomato spotted wilt virus (Mutschler, unpublished). Theoretically, the same could be accomplished with TYLCV resistance genes

## Figures and Tables

**Figure 1 insects-11-00842-f001:**
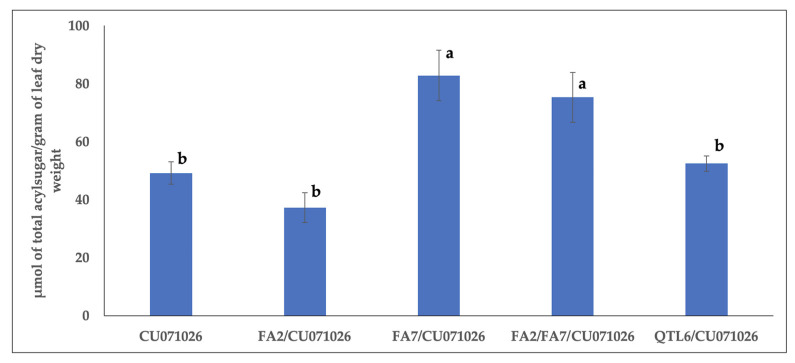
Bars represent averages of total acylsugars’ amount with standard errors in acylsugar-producing tomato lines in comparison with the benchmark breeding line CU071026. Bars with different letters indicate differences between treatment means using the Tukey–Kramer method following analysis of variance.

**Figure 2 insects-11-00842-f002:**
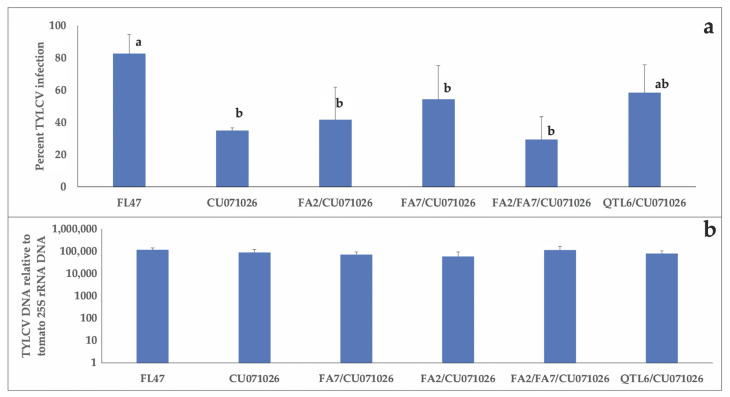
(**a**) Tomato plants representing each acylsugar-producing line and Florida 47 (FL47) were individually inoculated using twenty viruliferous whiteflies. Three weeks post inoculation, the infection status of each plant was assessed via endpoint PCR. Treatment means were separated using the Tukey–Kramer method following analysis of variance. Different letters indicate differences among treatments. (**b**) Averages of tomato yellow leaf curl virus (TYLCV) DNA in relation to the tomato housekeeping gene (25SrRNA) DNA with standard errors in acylsugar-producing tomato lines in comparison with FL47 following whitefly-mediated inoculation. DNA extracted from TYLCV positive plant samples (n = 7 to 19/treatment) as determined by endpoint PCR alone were used for quantitative PCR. Treatment means were separated using the Tukey–Kramer method following analysis of variance.

**Figure 3 insects-11-00842-f003:**
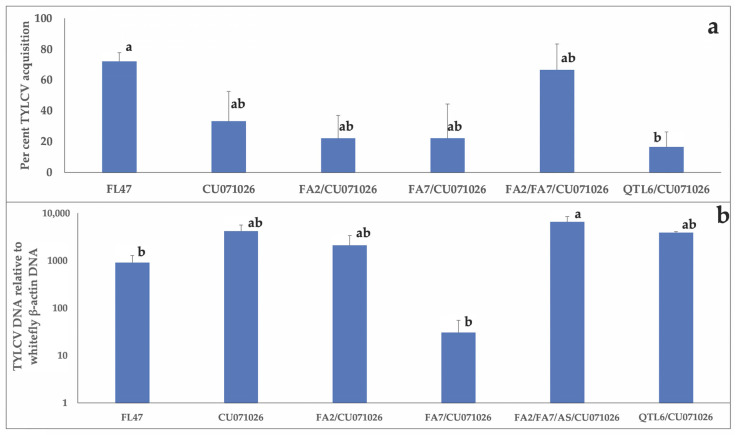
(**a**) Whiteflies clip-caged on tomato leaflets of acylsugar-producing lines and FL47 for 72 h were individually tested for the presence or absence of TYLCV using endpoint PCR. The treatment means were separated using the Tukey–Kramer method following analysis of variance. Different letters within the column indicate differences among treatments. (**b**) Bars represent averages of TYLCV loads with standard errors in relation to the whitefly housekeeping gene (β-actin) DNA in individual whiteflies following acquisition from acylsugar-producing lines and FL47 DNA extracted from TYLCV positive whitefly samples (n = 3 to 13/treatment) as determined by endpoint PCR alone were used for quantitative PCR. The treatment means were separated using the Tukey–Kramer method following analysis of variance. Different letters within the column indicate differences among treatments.

**Figure 4 insects-11-00842-f004:**
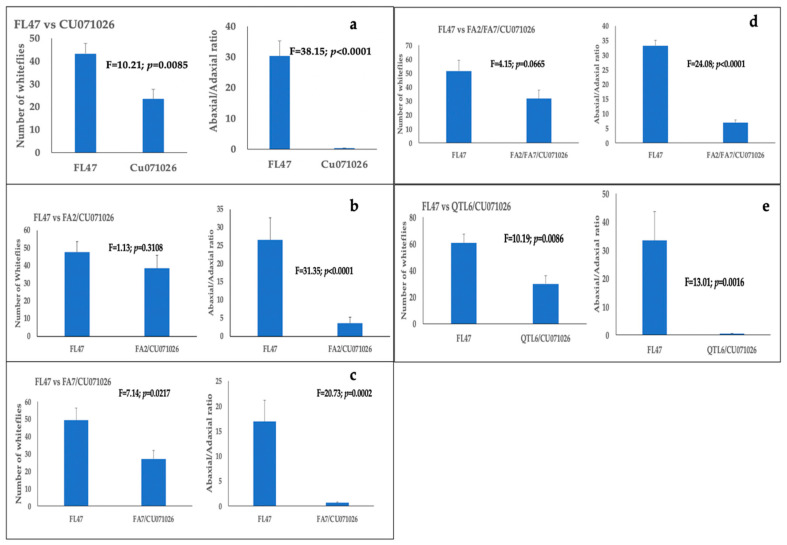
Bars with standard errors represent whitefly settling differences between hybrid FL47 and five acylsugar-producing tomato lines: (**a**) FL47 vs. CU071026; (**b**) FL47 vs. FA2/CU071026; (**c**) FL47 vs. FA7/CU071026; (**d**) FL47 vs. FA2/FA7/CU071026; and (**e**) FL47 vs. QTL6/CU071026. Bars with standard errors also represent ratios of whiteflies settling on adaxial and abaxial leaf surfaces of each leaf placed in the settling arena. Whitefly counts were made 24 h after release.

**Figure 5 insects-11-00842-f005:**
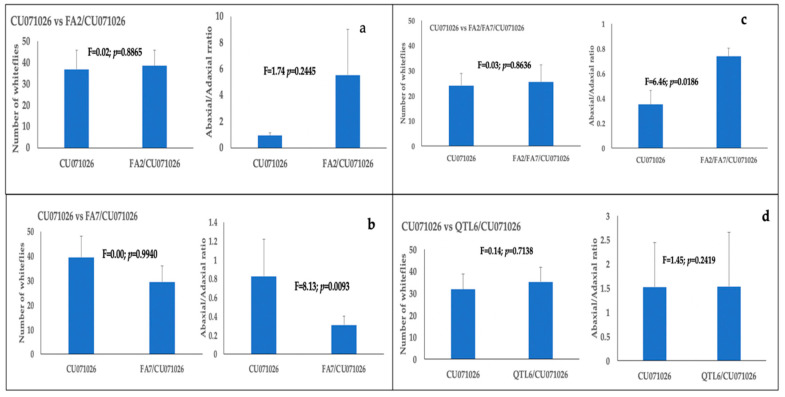
Bars with standard errors represent whitefly settling differences between CU071026 and four other acylsugar-producing lines: (**a**) CU071026 vs. FA2/CU071026; (**b**) CU071026 vs. FA7/CU071026; (**c**) CU071026 vs. FA2/FA7/CU071026; and (**d**) CU071026 vs. QTL6/CU071026. Bars with standard errors also represent ratios of whiteflies settling on adaxial and abaxial leaf surfaces of each leaf placed in the settling arena. Whitefly counts were made 24 h after release.

**Table 1 insects-11-00842-t001:** Composition of fatty acids from five acylsugar-producing lines.

	Fatty Acids Type														
Genotypes	i-C4	ai-C5	i-C5	ai-C6	i-C6	i-C9	i-C10	n-C10	i-C11	ai-C11	n-C11	i-C12	n-C12	i-C13	i-C14
CU071026	0.65b	19.65b	60.92a	0.08ab	0.01a	0.00b	0.00b	0.43c	0.00c	0.00b	0.00a	0.00a	17.71b	0.00c	0.56b
FA2/CU071026	3.50a	26.75a	56.77a	0.00b	0.00a	0.02b	0.02b	0.02c	5.22b	0.00b	0.00a	0.02a	3.60d	2.27a	1.82a
FA7/CU071026	0.32b	19.58b	49.82b	0.13ab	0.02a	0.00b	0.00b	7.79a	0.05c	0.00b	0.01a	0.00a	21.86a	0.01c	0.41b
FA2/FA7/CU071026	0.36b	19.05b	44.51b	0.08ab	0.01a	0.51a	0.28a	6.09b	8.68a	0.31a	0.01a	0.01a	19.39ab	0.23b	0.48b
QTL6/CU071026	0.68b	26.76a	59.19a	0.09ab	0.00a	0.00b	0.00b	0.32c	0.00c	0.00b	0.00a	0.00a	12.45c	0.00c	0.50b

Average percentages of fatty acids extracted from the sugar backbone of various acylsugar-producing lines are presented. Seven foliar samples representing replications in duplicates were used for this estimation. Treatment means were separated using the Tukey–Kramer method following analysis of variance. Different letters within each column indicate differences among treatments.

**Table 2 insects-11-00842-t002:** Whiteflies’ survival from egg to late instar immatures.

Genotype	Percentage + Standard Error ^a^
FL47	90.7 ± 4.86a
CU071026	47.6 ± 2.64b
FA2/FA7/CU071026	49.1 ± 11.79b
QTL6/CU071026	61.5 ± 11.93b

^a^ Whitefly survival was measured on FL47 and three acylsugar-producing lines by allowing adult females to oviposit on the abaxial leaf surface; following which, eggs were monitored until they matured into third or fourth instar nymphs in approximately two weeks post oviposition. The treatment means were separated using the Tukey–Kramer method following analysis of variance. Different letters within the column indicate differences among treatments.

**Table 3 insects-11-00842-t003:** Whiteflies’ developmental time from egg to adult eclosion in a non-acylsugar hybrid in relation to acylsugar-producing lines.

Genotype ^a^	Median ^b^	N	Sum of Scores	Expected Under H0	Std Dev Under H0	Mean Score
FL47	27 (22–35)	34	27.33	17.00	2.32	0.80
CU071026	25 (18–31)	41	15.33	20.50	2.44	0.37
FA2/FA7/CU071026	24 (19–31)	14	2.33	7.00	1.65	0.17
QTL6/CU071026	25 (19–27)	33	16.00	16.50	2.30	0.48

^a^ Whitefly median developmental time was measured from egg to adult on FL47 and three acylsugar-producing lines at 24 h time intervals using a clip cage affixed to the abaxial leaf surface. ^b^ Median developmental time along with range on both FL47 and acylsugar-producing lines.

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
