# Peer review of "Evaluating Acylsugars-Mediated Resistance in Tomato against *Bemisia tabaci* and Transmission of Tomato Yellow Leaf Curl Virus"

_insects, 2020, doi:10.3390/insects11120842_

Round 1
Reviewer 1 Report
The authors characterize tomato lines producing acylsugars. In previous work they tested them to deter other insects and diseases; here they report on their effect on whiteflies and TYLCV. Compared to non acylsugar producing tomato plants, the lines accumulating various acylsugars and fatty acids to various levels, decrease virus inoculation during 24 h inoculation access period and lower virus acquisition during 72 h acquisition access periods. The effect does not correlate with acylsugar quantity or composition. The viral titre in infected plants is unchanged. In settling experiments, whiteflies prefer an acylsugar-less line over acylsugar-producing lines and prefer abaxial leaf sides over adaxial leaf sides, especially for the non-acylsugar producing control. Trichome density might be a possible cause, but this is not tested. Whiteflies on acylsugar producing plants survive less and reach eclosion faster, indicative of negative impact of the acylsugars on whitefly performance.
The experiments and conclusions are sound and extend, as stated above, previous results obtained with these plants.
I have some minor points that the authors should consider:
I did not find the number of plants analyzed in figure 2b. Please provide the number.
Figure 4: In some experiments there seems to be a significant number of whiteflies that do not settle on any leaf. Please explain or complement the data. Figure 4b: can it be that probabilities are switched between number of whiteflies and ratio?).
Concerning the survival experiments, you tested only three of the five acylsugar producing lines. Please explain.
Line 160: repetition with line 168 (seven leaves).
Line 212: Just a note. You seem to have quantified TYLCV in whiteflies without purging the insects. So you will detect internalized virus and virus still present in the gut lumen.
Author Response
My co-authors and I appreciate the reviewer’s comments very much, we sincerely that believe that it has contributed to the improvement of the manuscript. We have carefully considered each one and have addressed each to the best of our abilities. Our replies to the reviewer’s comment/question is directly incorporated below each comment in bold. Thank you!
The authors characterize tomato lines producing acylsugars. In previous work they tested them to deter other insects and diseases; here they report on their effect on whiteflies and TYLCV. Compared to non acylsugar producing tomato plants, the lines accumulating various acylsugars and fatty acids to various levels, decrease virus inoculation during 24 h inoculation access period and lower virus acquisition during 72 h acquisition access periods. The effect does not correlate with acylsugar quantity or composition. The viral titre in infected plants is unchanged. In settling experiments, whiteflies prefer an acylsugar-less line over acylsugar-producing lines and prefer abaxial leaf sides over adaxial leaf sides, especially for the non-acylsugar producing control. Trichome density might be a possible cause, but this is not tested. Whiteflies on acylsugar producing plants survive less and reach eclosion faster, indicative of negative impact of the acylsugars on whitefly performance.
The experiments and conclusions are sound and extend, as stated above, previous results obtained with these plants.
I have some minor points that the authors should consider:
I did not find the number of plants analyzed in figure 2b. Please provide the number.
Fig 2b. This information is currently provided “DNA extracted from positive plant samples (n=7 to 19/treatment) as determined by endpoint PCR alone were used for quantitative PCR."
This information is also provided for 3b. "DNA extracted from TYLCV positive whitefly samples (n=3 to 13/treatment) as determined by endpoint PCR alone were used for quantitative PCR."
Figure 4: In some experiments there seems to be a significant number of whiteflies that do not settle on any leaf. Please explain or complement the data. Figure 4b: can it be that probabilities are switched between number of whiteflies and ratio?).
The whitefly settling percentages ranged from 66. 57 to 90.75 percent among treatment pairs. This information has now been acknowledged in the results section (lines 329-330). In this series of assays, settling was compared between non-acylsugar hybrid FL47 with acylsugar lines, the settling on FL47 was more or less similar 43 to 51% except in one instance where it was 60%. The variation was probably induced by settling on acylsugar-producing lines, and in at least one instance the overall settling was less. It is not clear what prompted this reduction.
The settling percentages were actually lower when only acylsugar-producing lines were used as treatment pairs. The whitefly settling percentages in assays involving acylsugar-producing lines alone ranged from 49.58 to 75.49% across treatment pairs. This information also has now been acknowledged in the results section (lines 342-343).
The whiteflies were reared on cotton prior to using them on these assays. Perhaps, the host switch could have contributed to some insects just settling on the experimental arena instead of the tomato leaves. This probably contributed to the overall reduction in settling percentages.
Despite this minor issue, the settling trend has been pretty consistent across replications within an experiment and/or experimental repeats. This experiment involved six replications and was repeated once more (n=12 for each settling assay pair). The results indicate a good consistency across replications within an experiment and within two experimental repeats.
The probabilities were switched in figure 4b, thanks so much for pointing that out. We have now corrected the issue.
Concerning the survival experiments, you tested only three of the five acylsugar producing lines. Please explain.
That is true. Poor germination of FA2 and FA7 seeds in this instance and time constraints coerced us to go ahead with the remaining acylsugar-producing lines. Since both FA2 and FA7 backgrounds were in the FA2/FA7 line, we decided to forego the FA2 and FA7 lines for this experiment. Despite their non-inclusion, the trend on negative fitness effects induced by acylsugar-producing lines on whiteflies is evident and consistent.
Line 160: repetition with line 168 (seven leaves).
Repetition has now been deleted.
Line 212: Just a note. You seem to have quantified TYLCV in whiteflies without purging the insects. So you will detect internalized virus and virus still present in the gut lumen.
This is a very good point. We provided whiteflies with an acquisition access period of 72h on all genotypes and that should have allowed sufficient amount of TYLCV to reach the salivary glands. Nevertheless, as the reviewer describes, we could have overestimated the amount of TYLCV not directly associated with inoculation from salivary glands. We could have avoided this situation by purging TYLCV in whiteflies by placing them on a non-TYLCV host plant. We will keep this useful tip in mind for future studies.
Reviewer 2 Report
Specific comments: I believe this is a fantastic paper, but the Abstract needs work.
Line 51 over acylsugar (producing?) lines
Line 51-56: The major findings of the study are mentioned. While I am certain the article will do this, it seems strange the authors chose a qualitative description for all of these results in the abstract. My first reaction was, they do not have any statistical treatments to support their findings? Example: “developmental time of whiteflies was shorter on acylsugar-producing lines compared to FL47” to whitefly developmental time in hours was 2 fold shorter on acylsugar-producing lines compared to FL47” Line 55: What is the criteria for “Whiteflies transmitted TYLCV?” Counting of necrotic regions, immunohistochemistry, in situ PCR? I do think that the readability is important, however your results are stunning.
Line 117 Secondary spread of the virus was also reduced but not eliminated in the acylsugar-producing genotype. It would be helpful to clarify what is defined as “secondary spread” and how it differs from primary spread.
Line 133 ….TYLCV to acylsugar-producing lines was assessed both qualitatively and quantitatively via PCR (bands +/-. End point quantitative PCR? Please clarify. (in the M&M it is clear, please address).
Table 3: Very interesting data. Also statistics first rate. 1-2 days longer (of about 25 days) is an enormous delay. Very nice experimental design.
Bottom Line
This study reflects a strong commitment by the authors to get to the bottom of one of the most vexing problems in agriculture. Pests + viruses that they carry vs acylsugars wonderfully portrayed.
Suggestions: The simple abstract is not bad, however the actual abstract is so general and uninspiring I would probably ask someone else to take the lead: Preferably someone familiar with the methodologies and design. Perhaps the idea was to create an Abstract that was not detailed, however I believe the result was a disservice to the research reported here. I would have not read the paper with this Abstract.
Author Response
My co-authors and I appreciate the reviewer’s comments very much, we sincerely that believe that it has contributed to the improvement of the manuscript. We have carefully considered each one and have addressed each to the best of our abilities. Our replies to the reviwer’s comment/question is directly incorporated below each comment in bold. Thank you!
Specific comments: I believe this is a fantastic paper, but the Abstract needs work.
The abstract has been completely modified based on the reviewer’s comments, we believe that they now more clearly describe the outcomes of the experiments in this study.
Line 51 over acylsugar (producing?) lines
You are correct. It should have been acylsugar-producing. Has now been modified.
Line 51-56: The major findings of the study are mentioned. While I am certain the article will do this, it seems strange the authors chose a qualitative description for all of these results in the abstract. My first reaction was, they do not have any statistical treatments to support their findings? Example: “developmental time of whiteflies was shorter on acylsugar-producing lines compared to FL47” to whitefly developmental time in hours was 2 fold shorter on acylsugar-producing lines compared to FL47” Line 55: What is the criteria for “Whiteflies transmitted TYLCV?” Counting of necrotic regions, immunohistochemistry, in situ PCR? I do think that the readability is important, however your results are stunning.
We appreciate the reviewer’s comments. We have thoroughly revamped the abstract with both qualitative and quantitative description of our results. The generic aspects have been removed and are replaced with outcomes of the experiments from this study. We believe that the current abstract more clearly represents the outcomes observed in this study.
Line 117 Secondary spread of the virus was also reduced but not eliminated in the acylsugar-producing genotype. It would be helpful to clarify what is defined as “secondary spread” and how it differs from primary spread.
Secondary spread of TYLCV following acquisition by whiteflies from TYLCV-infected acylsugar-producing ABL14-8 and inoculation of the non-acylsugar producing cultivar was also reduced but not eliminated [Reference 58]. Currently in lines 117-120.
Line 133 ….TYLCV to acylsugar-producing lines was assessed both qualitatively and quantitatively via PCR (bands +/-. End point quantitative PCR? Please clarify. (in the M&M it is clear, please address).
This sentence has now been modified as “the ability of whiteflies to inoculate TYLCV to acylsugar-producing lines was assessed both qualitatively and quantitatively via endpoint PCR and quantitative PCR, respectively.” Currently in lines 134-135.
Table 3: Very interesting data. Also statistics first rate. 1-2 days longer (of about 25 days) is an enormous delay. Very nice experimental design.
Bottom Line
This study reflects a strong commitment by the authors to get to the bottom of one of the most vexing problems in agriculture. Pests + viruses that they carry vs acylsugars wonderfully portrayed.
Suggestions: The simple abstract is not bad, however the actual abstract is so general and uninspiring I would probably ask someone else to take the lead: Preferably someone familiar with the methodologies and design. Perhaps the idea was to create an Abstract that was not detailed, however I believe the result was a disservice to the research reported here. I would have not read the paper with this Abstract.
We appreciate the reviewer’s comments. We have thoroughly rewritten the abstract with both qualitative and quantitative description of our results. The generic aspects have been removed and are replaced with outcomes of the experiments from this study. While doing so, we have also attempted to conform to the 300-word limit. We believe that the current abstract presents a clearer reflection of the outcomes observed in this study.